# Thermal Conductivity Enhancement of Atomic Layer Deposition Surface-Modified Carbon Nanosphere and Carbon Nanopowder Nanofluids

**DOI:** 10.3390/nano12132226

**Published:** 2022-06-29

**Authors:** Marcell Bohus, Thong Le Ba, Klara Hernadi, Gyula Gróf, Zoltán Kónya, Zoltán Erdélyi, Bence Parditka, Tamás Igricz, Imre Miklós Szilágyi

**Affiliations:** 1Department of Inorganic and Analytical Chemistry, Budapest University of Technology and Economics, Muegyetem Rakpart 3, 1111 Budapest, Hungary; kenty9x@gmail.com; 2Institute of Physical Metallurgy, Metal Forming and Nanotechnology, University of Miskolc, 3515 Miskolc-Egyetemváros, Hungary; femhernadi@uni-miskolc.hu; 3Centre for Energy Research, Konkoly-Thege Miklós út 29-33, 1121 Budapest, Hungary; grof.gyula@ek-cer.hu; 4Department of Applied and Environmental Chemistry, University of Szeged, Rerrich Béla tér 1, 6720 Szeged, Hungary; konya@chem.u-szeged.hu; 5Department of Solid State Physics, Faculty of Science and Technology, University of Debrecen, P.O. Box 400, 4002 Debrecen, Hungary; zoltan.erdelyi@science.unideb.hu (Z.E.); parditka.bence@science.unideb.hu (B.P.); 6Department of Organic Chemistry and Technology, Budapest University of Technology and Economics, Budafoki út 8, 1111 Budapest, Hungary; igricz.tamas@vbk.bme.hu

**Keywords:** nanofluid, atomic layer deposition, thermal conductivity, viscosity, carbon nanosphere, carbon nanopowder, titanium dioxide

## Abstract

In this paper, we present a study on thermal conductivity and viscosity of nanofluids containing novel atomic layer deposition surface-modified carbon nanosphere (ALD-CNS) and carbon nanopowder (ALD-CNP) core-shell nanocomposites. The nanocomposites were produced by atomic layer deposition of amorphous TiO_2_. The nanostructures were characterised by scanning (SEM) and transmission electron microscopy (TEM), energy dispersive X-ray analysis (EDX), Fourier transform infrared spectroscopy (FT-IR), Raman spectroscopy, thermogravimetry/differential thermal analysis (TG/DTA) and X-ray powder diffraction (XRD). High-concentration, stable nanofluids were prepared with 1.5, 1.0 and 0.5 vol% nanoparticle content. The thermal conductivity and viscosity of the nanofluids were measured, and their stability was evaluated with Zeta potential measurements. The ALD-CNS enhanced the thermal conductivity of the 1:5 ethanol:water mixture by 4.6% with a 1.5 vol% concentration, and the viscosity increased by 37.5%. The ALD-CNS increased the thermal conductivity of ethylene–glycol by 10.8, whereas the viscosity increased by 15.9%. The use of a surfactant was unnecessary due to the ALD-deposited TiO_2_ layer.

## 1. Introduction

Maintaining efficient heat transfer is essential in many engineering processes and systems that serve our modern life. Heat transfer has three primary forms—conductivity, convection, and radiation—and all basic heat transfer phenomena have specific methodologies for describing the heat transfer rate calculated by engineering calculations. Although the conductivity, convection and radiation may appear in their pure form, calculation of heat transfer is more general than each of these forms individually. For example, in the overall heat transfer process, convection (+radiation)—conduction—convection (+radiation) occur as serial processes. In many engineering applications, convection is the weakest term in the heat transfer chain. Like the other basic forms of heat transfer, convection is a complex phenomenon. First, the energy (heat) must penetrate the flowing fluid, and only after that can the energy be transported. The penetration rate strongly depends on the stationary thermal conductivity of the fluid. The transport capacity of the fluid depends on its heat capacity, and the flow of the fluid is mainly characterised by the viscosity. The intensification of the convection may occur in various ways, but the low thermal conductivity of the generally used heat transfer fluid implies a specific limit for those techniques. This is the central issue that motivates engineers and scientists to produce more suitable heat transfer fluids not only for utilization with traditional energy sources but especially for solar energy applications.

Solar collectors require efficient ways to transport captured energy. For these purposes, conventional working fluids, such as water, as well as various kinds of glycols and oils, are used [1].

Increasing the thermal conductivity of the working fluids using dispersed nanostructures intensifies heat transport by increasing the heat penetration into the fluid. Most resulting nanofluids are made from a base fluid, most commonly water, ethylene glycol, mineral oil or a mixture of thereof, and different types of nanoparticles are dispersed in this base fluid [2]. The most studied nanostructures are metal-, metal-oxide- and carbon-based nanomaterials [3].

Thermal conductivity (λ) and dynamic viscosity (m) are two key properties studied with respect to nanofluids. According to the second law of thermodynamics, any increase in heat transfer must result in increased entropy generation. An increase in entropy generation manifests in increased pumping work required to maintain the working fluid flow when heat transfer is intensified. An increase in thermal conductivity intensifies heat transfer, whereas increased viscosity results in increased pumping work or increased pressure losses. With the addition of different nanoparticles, the various properties of the base fluid can be modified [4]. Filho et al. studied silver nanoparticle-based nanofluids in direct absorption solar collectors. The use of silver nanoparticles at low concentrations (1.62, 3.25 and 6.5 ppm) enhanced photothermal conversion [5]. Although the use of noble metals shows great potential, their price represents a barrier to their application. Affordable alternatives include oxide- and carbon-based nanomaterials, although oxides are not as efficient thermal conductors as carbon. Eastman et al. studied water-based copper oxide (36 nm) and aluminium oxide (33 nm) nanofluids. With 5 vol% copper-oxide, the thermal conductivity increased by 60% and 40% with aluminium oxide [6]. In 2001, the same authors studied ethylene–glycol-based copper nanofluids and observed a 40% increase in thermal conductivity with the use of only 0.3 vol% copper nanoparticles (<10 nm) [7]. Ravi Kumar et al. experimented with Fe_3_O_4_ nanofluids in U-bend heat exchangers with exchangeable strip inserts. According to the forced flow characteristics, the Nusselt number on the nanofluid side increased with increasing Reynolds number and nanoparticle concentration. The increase was 14.7% with 0.06% nanoparticles and 41.29% with the same concentration and aspect ratio = 1 longitudinal strip insert of the heat exchanger [8]. Duangthongsuk et al. studied TiO_2_ nanofluids and observed a 7.2% increase in thermal conductivity for 2.0 vol% nanoparticle concentration [9]. Prado et al. studied MgO dispersed in n-tetradecane. The thermal conductivity of the nanofluid rose by 17% with 10 wt% nanoparticle concentration compared to the base fluid [10]. Yoo et al. compared TiO_2_/water, Al_2_O_3_/water, Fe/ethylene–glycol and WO_3_/ethylene–glycol nanofluids. With a 1.0 vol% concentration, the TiO_2_ nanofluid exhibited a 14.4% enhancement in thermal conductivity, whereas the Al_2_O_3_ nanofluid exhibited a 4 % enhancement. Fe and WO_3_ nanofluids with 0.3 vol% concentrations exhibited a 16.5% and 13.8% increase in thermal conductivity, respectively [11]. The thermal conductivity of carbon nanostructures is significantly higher than that of metals and metal oxides. For example, copper oxide has a thermal conductivity of 77 W m^−1^ K^−1^, and copper has a 398 W m^−1^ K^−1^ thermal conductivity, whereas that of carbon nanotubes ranges from 2000 to 6000 W m^−1^ K^−1^, which is one order of magnitude higher [3].

Solar collectors absorb solar radiation and transform energy in the form of heat. The heat is conveyed by a work fluid, most commonly water, air or oil. The working fluid then transports the energy to a heat storage unit or directly to the required hot water [12]. There are several difficulties associated with direct absorption solar collectors, including low energy utilisation, inefficient heat transfer, heat loss and poor solar absorption properties. The heat transfer and solar absorption properties of working fluids can be optimised using carbon-nanostructure-based nanofluids [3]. There are various possible modifications that can be made to carbon with different properties, providing a large field for nanofluids research. Promising nanocarbon candidates include single- (SWCNT) and multi-wall carbon nanotubes (MWCNT), fullerenes, graphite, graphene and graphene oxide nanoparticles, carbon black and diamond. At lower concentrations, they afford entirely black nanofluids, making them suitable for application in direct absorption solar collectors. Struchalin et al. studied the performance of a tubular direct absorption solar collector using MWCNT nanofluids. With different concentrations, the performance increased by 5.8–37.9%, with an optimal concentration of 0.01 wt% compared to a surface absorption solar collector with similar geometry [13]. Li et al. experimented on a water-based, stable, high-concentration functionalised MWCNT nanofluid. They concluded that thermal conductivity increased by 32% and 69.7% for 0.5 and 2.5 vol% concentrations, respectively. On the contrary, due to the high concentrations, the viscosity rose by roughly 20 times to 31 mPas [14].

Besides simple nanofluids with only one kind of nanoparticle, there are hybrid nanofluids that contain two or more different nanoparticles. Additionally, the nanoparticles can contain two or more materials, producing nanocomposite nanofluids. One way to create pseudo-hybrid nanofluids is by using atomic layer deposition (ALD). ALD processes are commonly based on a binary, self-limiting reaction sequence. During an ALD sequence, the first precursor is bound up on the particle surface, and then the second precursor’s reaction with the first precursor creates the final chemical structure. With the cycle repeating, the layer thickness can be set with atomic accuracy [15].

Baghbanzadeh studied SiO_2_-MWCNT hybrid nanofluids. With 50–50% nanoparticle composition, the enhancement of the thermal conductivity was lower than what the ratio would suggest, as the SiO_2_ wedged between the MWCNT clusters and worked as an insulator between them [16].

Esfe et al. worked on the mathematical determination of different hybrid nanofluids (Ag-MgO/water, Cu-TiO_2_/water-ethylene–glycol (60:40) and SWCNT-MgO (20:80)/ethylene–glycol) with a maximum concentration of 2 vol%. Using an artificial neural network, they produced equations to determine the thermal conductivity of the hybrid nanofluids. As a result, the equations were non-linear with respect to the ratio of the nanoparticles [17,18,19].

Bakhtiari et al. studied graphene-TiO_2_ (30:70)/water nanofluids. The largest thermal conductivity increase was 27.84% with 0.5 vol% concentration at 75 °C [20]. Van Trinh et al. studied ethylene–glycol-based copper-decorated graphene-MWCNT hybrid nanofluids. The results showed that the nanofluids had good stability due to the surface functional groups of the nanomaterials, and they also managed to increase the thermal conductivity by 41% with 0.035 vol% concentration at 60 °C [21].

Application of the ALD technique in nanofluids has not been common in recent studies, and there are only a few studies regarding the use of ALD-prepared core-shell nanoparticles in nanofluids. The thermal, rheological, mechanical and chemical properties of nanomaterials can be affected by applying a surface layer with a thickness of a few nanometres. Gil-Font et al. used molecular layer deposition to stabilise oil-based tin nanofluids. They deposited a polyethene-terephthalate layer on the nanoparticles, as aided in particle dispersion in the base fluid [22]. Navarrete et al. experimented with Al_2_O_3_-coated Sn nanoparticles to increase the thermal energy storage capacity of molten salt in concentrated solar power plants [23].

Shang et al. studied the optical absorption properties and photothermal conversion properties of Al_2_O_3_-coated silver nanoparticles in Therminol 66 base fluid. The coating served as an anticorrosive layer on the silver core (45 nm) with a thickness of 2.7 nm. As a result, most of the solar radiation was absorbed in a 1.75 cm thin nanofluid layer with a concentration of 0.04 wt% [24].

In this paper, novel ALD-modified carbon-nanostructure-based nanofluids were studied with water/ethanol and ethylene–glycol base fluids, following previous research regarding the original carbon nanosphere (CNS) and carbon nanopowder (CNP) materials [25]. Several nanometres of TiO_2_ was grown on the surface of the nanoparticles according to the ALD process. The particles were analysed with FT-IR, Raman spectroscopy, TEM, SEM-EDX, XRD and TGA. The nanofluids were prepared in 0.5, 1.0 and 1.5 vol% concentrations with an ethanol–water (1:5) base fluid mixture for the CNS and ethylene–glycol for the CNP. Their thermal conductivity and viscosity were measured at temperatures of 20, 30, 40, 50 and 60 °C. The stability of the nanofluids was examined with Zeta potential measurements.

## 2. Materials and Methods

### 2.1. Materials

Carbon nanospheres were prepared from sugar using the hydrothermal method [26]. Watery sucrose solution with a pH of 12 with NaOH was treated in an autoclave for 12 h under autogenous pressure at 180 °C. The product was washed with distilled water three times and then suspended in a 45% ethanol–water solution, and then the suspension was centrifuged at 4000× *g* rpm for 20 min. The settled material was filtered and washed with warm, distilled water, then dried at 70 °C overnight. The product was a fine brown powder [27].

Carbon nanopowder (CAS 7440-44-0) and ethylene–glycol (CAS 107-21-1) were purchased from Sigma Aldrich. Ethyl–alcohol was purchased from GPR Rectapur (CAS 64-17-5).

ALD modification of the nanoparticles was performed using H_2_O and TiCl_4_ precursors in a Beneq (Espoo, Finland) TFS-200 reactor [28]. A total of 154 ALD cycles were repeated with 3000 ms N_2_ purge and 300 ms precursor pulses at 1.3 mbar pressure and 108 °C, which provided a few nanometres of amorphous surface TiO_2_.

### 2.2. Preparation of Nanofluids

Stable nanofluids were prepared from the original carbon nanospheres with the use of a 5:1 deionised water (DI)/ethanol base fluid. The original carbon nanopowder was stable in a 1:1 ethylene–glycol/DI mixture with surfactants [25]. After ALD, it was possible to avoid the use of surfactants. Hence, ALD-modified carbon nanopowder (ALD-CNP) nanofluids were prepared with ethylene–glycol base fluid without surfactants due to the ALD treatment. The ALD-modified carbon nanosphere (ALD-CNS) nanofluids were prepared with the original mixture of ethanol and DI. The prepared concentrations were 1.5, 1.0 and 0.5 vol%. The nanofluids were treated with ultrasound for 1h at 130 W and 45 kHz. The nanofluids were stable for several days.

### 2.3. Characterisation Techniques

The morphology of the nanoparticles was studied using a LEO 1440 XB SEM (LEOGmbH, Oberkochen, Germany) at 5 kV and a Philips CM20 TEM (Philips, Amsterdam, The Netherlands) at 200 kV. The elemental composition of the nanoparticles was examined by EDX analysis with a JEOL JSM-5500LV SEM (Tokyo, Japan). FT-IR spectroscopy was performed with an Excalibur FTS 3000 BioRad FT-IR (Bio-Rad, Hercules, California, USA)) with 4 cm^−1^ resolution and 64 accumulations in the range of 4000–600 cm^−1^ in KBr pastille. The crystallinity of the nanoparticles was determined using a PANanalytical X’pert Pro MPD XRD (Malvern Panalytical Ltd., Malvern, UK) under Cu_α_ irradiation at 40 kV and 30 mA. The measuring program consisted of 3°/min from 5 to 65 2Θ.

Raman spectroscopy experiments were performed with a HORIBA Jobin Yvon Raman spectrometer (Kyoto, Japan) at a wavelength of 532 nm (frequency-doubled Nd-YAG laser).

Thermal analysis of the nanoparticles was performed with an STD 2960 TG/DTA (TA Instruments Inc., New Castle, DE, USA). The experimental parameters for the ALD-CNS were from room temperature (RT) to 200 °C at a rate of 15 °C/min and 5 °C/min to 700 °C. For the ALD-CNP, the temperature was increased from RT to 600 °C at a rate of 10 °C/min. The experiments were conducted in an air atmosphere with a 130 mL/min flow rate.

The viscosity of the nanofluid was examined using an Anton Paar Physica MCR 301 rotation viscometer (Anton Paar, Ashland, VA, USA) at 20, 30, 40, 50 and 60 °C.

The thermal conductivity of the nanofluids was determined with an SKZ1061C TPS thermal conductivity tester (SKZ Industrial, Shandong, China). The thermal conductivity was measured at 20, 30, 40, 50 and 60 °C with three repetitive measurements. Zeta potential was measured with a Brookhaven ZETAPALS zeta potential measuring device (Brookhaven Instruments, New York, NY, USA).

The density of the nanoparticles was determined by measuring the water displacement of nanoparticles in a measuring flask.

## 3. Results and Discussion

### 3.1. Characterisation of Nanoparticles

Figure 1 shows SEM (a, e) and TEM (b, c, d) images of ALD-CNS (a, b) and ALD-CNP (c, d) nanoparticles, as well as the CNP, before ALD treatment (e). The ALD-CNS nanoparticles have a smooth surface and are mostly sphere-shaped. There are stuck particles, and the material has a large particle size distribution. The ALD-CNP nanoparticles are more uniform and have a smoothed surface compared to the original CNP. The average particle size of the CNS is 498 nm, as obtained in previous research by treatment of SEM images with respect to the original nanoparticles, whereas the CNP has an average diameter of 100 nm, according to the manufacturer [25]. In Figure 1c,d, the surface layer of the TiO_2_ can be observed with a thickness of 3 nm. In both nanomaterials, the particles form aggregated clusters, which must be broken down to increase the stability of the nanofluids.

The EDX measurements show 2.7 at% of TiO_2_ on the ALD-CNS and 0.8 at% on the ALD-CNP, which confirms the success of the ALD process.

The FT-IR spectra of CNS and CNP nanoparticles are shown in Figure 2 before and after ALD modification. In the cases of both nanomaterials, the characteristic ν (–OH) vibration band can be observed at ~3450 cm^−1^. The 1385 cm^−1^ peaks are caused by the δ (C–H) vibrations, which are absent in the case of the unmodified CNS. The peaks at 1633 and 1636 cm^−1^ are assigned to the ν (C=C) double-bond vibrations. The 1027 and 2360 cm^−1^ peaks are caused by the carbon dioxide in the air. For the ALD-CNS, the peak at 2932 cm^−1^ is assigned to the ν (C–H) vibration, whereas the peak at 1705 cm^−1^ is assigned to the ν (C=O) double-bond vibration. For the ALD-CNP, the peak at 2918 cm^−1^ is also assigned to the ν(C–H) vibration. The 800 cm^−1^ peaks are assigned to the aromatic C–H wagging motion [29].

Due to the relatively low concentrations, the characteristic absorptions of the TiO_2_ (3450, 1630 and <1000 cm^−1^) cannot be seen in the spectra, and they may overlap with the signals of the carbon nanostructures [30,31].

Figure 3 shows the Raman spectra of the ALD-CNS (a) and ALD-CNP (b) nanomaterials. For the ALD-CNS, first, the G and D bands of the CNS core can be observed, but with the use of longer laser exposure, the characteristic bands of the rutile and anatase forms of the TiO_2_ are observed due to the applied heat. For the ALD-CNP only G and D bands appeared, and the TiO_2_ bands could not be developed with the applied laser light. The G/D ratio for the ALD-CNS was 2.31 and 1.22 for the ALD-CNP, which means that there are many sp^3^ hybrid-state C atoms, with more structural defects compared to ALD-CNS.

The thermal analysis curves of the nanoparticles are presented in Figure 4, as the nanofluids are often used at elevated temperatures in contact with air. Additionally, the ALD process is performed at higher temperatures, which the nanostructures must withstand. The thermal decomposition of the ALD-CNS happens in two stages. In the first stage, the sample lost the absorbed and chemisorbed water and surface functional groups at 245 °C with a 6.30% mass loss. In the second stage, the slow exothermic burning of the sample started and accelerated drastically at around 450 °C. For the ALD-CNP, the water loss and decomposition of surface functional groups occurred at 425 °C, with a loss of only 4.00% of the sample mass. During the first stage, in the 200–320 °C range, an exothermic peak was observed, which was caused by the anatase transformation of the surface TiO_2_. After the first stage, slow exothermic burning started at a constant rate until near-complete decomposition at 650 °C. The ALD-CNS proved to be stable up to 200–250 °C, whereas the ALD-CNP was stable below 400 °C, leading to the conclusion that the tested nanofluids are safe to use in watery and ethylene–glycolic environments.

The XRD patterns of the nanoparticles are shown in Figure 5. There is no recognisable pattern in the XRD image of the ALD-CNS, whereas two wide bands are observed in the case of ALD-CNP. The 25.0°and 43.8° 2Θ bands are assigned to the (002) and (101) planes of graphite. The bands of crystalline TiO_2_ cannot be observed, as the deposited TiO_2_ was amorphous. Additionally, its most intensive diffraction peak would overlap with the 25 °C peak in the case of ALD-CNP.

### 3.2. Properties of Nanofluids

#### 3.2.1. Stability of the Nanofluids

The prepared ALD-CNS nanofluids had a zeta potential of −34.68, −38.21 and −39.88 mV, whereas the ALD-CNP nanofluids had a zeta potential of 35.61, 46.49 and 37.41 mV for 0.5, 1.0 and 1.5 vol% nanofluids, respectively, which means that the nanofluids can be considered stable.

#### 3.2.2. Viscosity of Nanofluids

The nanofluids behaved as Newtonian fluids as the shear stress rose linearly with the shear rate at a given temperature. Figure 6 shows the relative viscosity of the nanofluids compared to the base fluid. For the ALD-CNS nanofluids, the viscosity rose by 15.2 (50 °C), 25.9 (60 °C) and 37.5% (60 °C) maximum for 0.5, 1.0 and 1.5 vol% concentrations, respectively. The high-viscosity increase is explained by the large diameter and the increased density of the nanoparticles due to the ALD process. Another factor responsible for the viscosity increase is the fact that during the ALD process, particles can cohere in the deposited layer and have an increased impact cross section and distorted sphericity. For the ALD-CNP nanofluids, the viscosity rose by 5.0 (50 °C), 9.0 (40 °C) and 15.9% (40 °C) maximum for 0.5, 1.0 and 1.5 vol% concentrations, respectively. In this case, the viscosity did not increase as drastically as for the ALD-CNS, but the enhancement shows similar tendencies and can be explained similarly. The main differences are the smaller diameter of the ALD-CNP and the smaller density increases associated with ALD. The density of the ALD-CNS increased from 1.0 to 2.19 g/mL, whereas the density of the ALD-CNP rose from 1.8–2.1 to 2.24 g/mL.

In our previous research, the highest relative viscosity increase was 9.31 % for the CNS and 9.56 % for the CNP nanofluids. In comparison, the ALD-CNS nanofluids experienced a drastic increase of 37.5 %, whereas the ALD-CNP nanofluids exhibited a maximal viscosity increase of just 15.9 % [25].

In Figure 7, the dynamic viscosity is represented at different temperatures. Viscosity is known to be strongly dependent on temperature, and this is visualised by an exponential curve fit to the measurement points. The curve equation can be expressed as follows:$$y=a\;e^{-\frac{T}{b}}+C$$
where *a*, *b* and *C* are constants, and T is the temperature in °C. The parameters are summarised in Table 1.

#### 3.2.3. Thermal Conductivity of Nanofluids

The enhancement of the thermal conductivity (λ) of the base fluids is represented in Figure 8, and the absolute thermal conductivity of the nanofluids compared to the base fluids is shown in Figure 9. The enhancement is highly affected by the nanomaterial concentration in both cases. For the ALD-CNS, thermal conductivity was increased by 4.6 % maximum at a 1.5 vol% concentration at 30 °C. Compared to the original CNS nanofluids, the enhancement increased by four times [25]. In the case of the ALD-CNP nanofluids, the thermal conductivity enhancement exhibits a temperature-dependent tendency, which is caused by the increasing Brownian motion of the nanoparticles, which is more relevant than in the case of the ALD-CNS due to the smaller particle size of the ALD-CNP. Additionally, during the ALD process, the rough surface of the CNS particles became smooth, which can result in a smaller impact cross section and increased sphericity, resulting in higher particle mobility. The maximum enhancement was 10.8% at 60 °C with a 1.5 vol% concentration. Additionally, the higher thermal conductivity enhancement can be explained by the higher crystallinity of the ALD-CNP. Regarding the original CNP nanofluids, the thermal conductivity increase was more concentration-dependent, whereas temperature had little to no effect, contrary to the ALD-modified nanoparticles [25].

Figure 9 shows fitted lines corresponding to calculation of the thermal conductivity of any temperature covered by the measurements. The fitting parameters are shown in Table 2. Thermal conductivity shows a linear dependence on temperature with good fitting parameters.

Comparing the ALD-CNS and ALD-CNP nanofluids, the ALD-CNP is capable of higher thermal conductivity enhancement with increasing temperature, whereas the ALD-CNS nanofluids exhibited near-constant enhancement values in the examined temperature range. In the case of the ALD-CNS, lower temperature utilisation is advised, as it contains a significant amount of alcohol with higher absolute thermal conductivity than the ALD-CNP nanofluids. On the contrary, ALD-CNP nanofluids show a clear tendency to increase thermal conductivity enhancement, which is favourable at higher temperatures, even near 100 °C. The use of higher concentrations is advised, but the viscosity increase cannot be forgotten, as the concentration has an even greater impact.

Table 3 shows the comparison between our nanofluids and other nanofluids reported in the literature. There are mixed results in the presented data. Unfortunately, viscosity measurements are not available in most cases. Overall, our nanofluids exhibit an average thermal conductivity increase compared to the values reported in other studies. In the case of viscosity, high concentrations of solid additives tend to cause a high-viscosity increase, which is confirmed by the viscosity measurements.

## 4. Conclusions

A few nanometres of titanium dioxide was successfully grown on the surface of CNS and CNP nanoparticles using ALD. The deposition was confirmed with the use of SEM-EDX and TEM measurements. Characterisation of the nanoparticles was performed using SEM-EDX, TEM, FT-IR, XRD, TG-DTA and Raman spectroscopy measurements.

The various concentrations of ALD-CNS and ALD-CNP nanofluids were successfully prepared in 1/5 ethanol:DI and ethylene–glycol base fluids. The nanofluids were stable according to zeta potential measurements. The highest thermal conductivity increase compared to the applied base fluids was 4.6% for the ALD-CNS and 10.8% for the ALD-CNP. The relative viscosity of the nanofluids was 37.5% and 15.9% for the ALD-CNS and ALD-CNP, respectively.

The thermal conductivity enhancement of the ALD-CNS is more favourable at lower temperatures, whereas the ALD-CNP has more potential for application at higher temperatures. Additionally, the relative viscosity of the ALD-CNS is significantly higher.

The significant difference compared to the unmodified CNS nanofluids can be attributed to the fact that stable nanofluids were prepared without surfactant with ALD application. Additionally, the thermal conductivity enhancement was several times higher due to the additional TiO_2_ layer.

The presented research is a pioneer study with respect to the application and characterisation of ALD composite nanofluids and is believed to be a valuable source of information for simulations in complex hydrodynamic systems.

The prepared nanofluids show high potential for use as a working fluid for direct absorption solar collectors. The nanofluids, especially the ALD-CNS nanofluids, have high light absorption capabilities and are suspected to induce high solar–thermal conversion efficiency. With direct solar absorption, the major drawback of surface absorption solar collectors can be bypassed, as heat transport is omitted between the pipe and the working fluid. Therefore, the pipe should be built from a transparent heat-insulating material, decreasing the heat loss.

## Figures and Tables

**Figure 1 nanomaterials-12-02226-f001:**
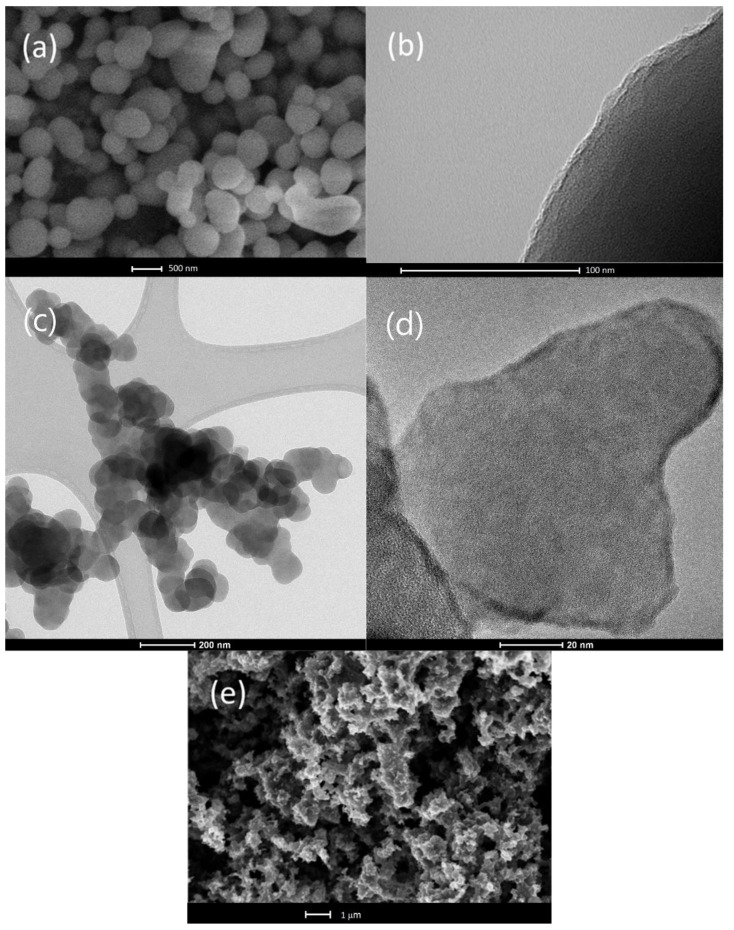
SEM image of (**a**) ALD-CNS with 50,000× magnification, TEM images of (**b**) ALD-CNS and (**c**,**d**) ALD-CNP and SEM image of (**e**) the original CNS with 20,000× magnification.

**Figure 2 nanomaterials-12-02226-f002:**
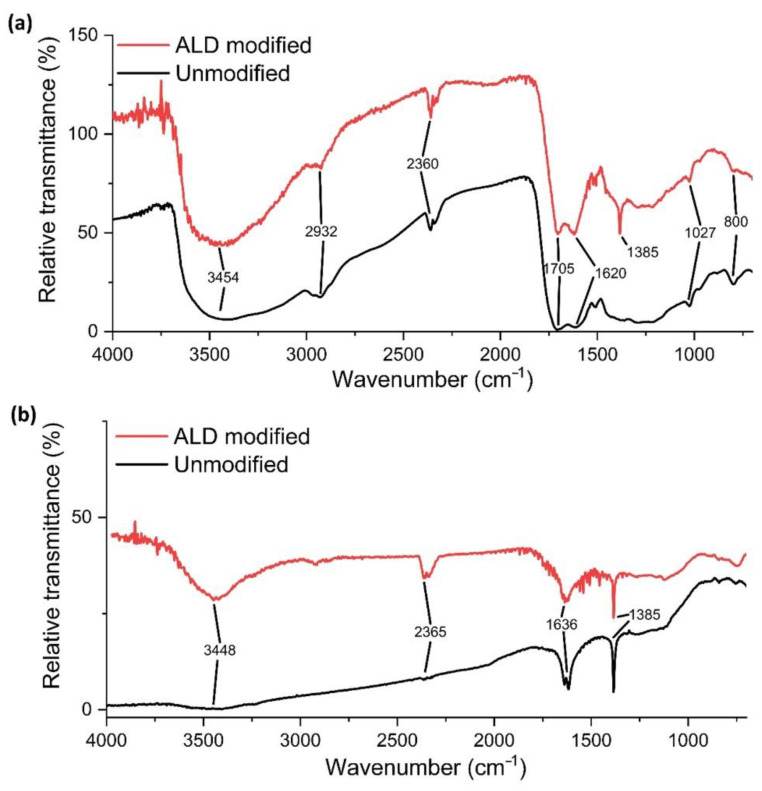
FT-IR spectra of (**a**) carbon nanospheres and (**b**) carbon nanopowder before and after ALD treatment.

**Figure 3 nanomaterials-12-02226-f003:**
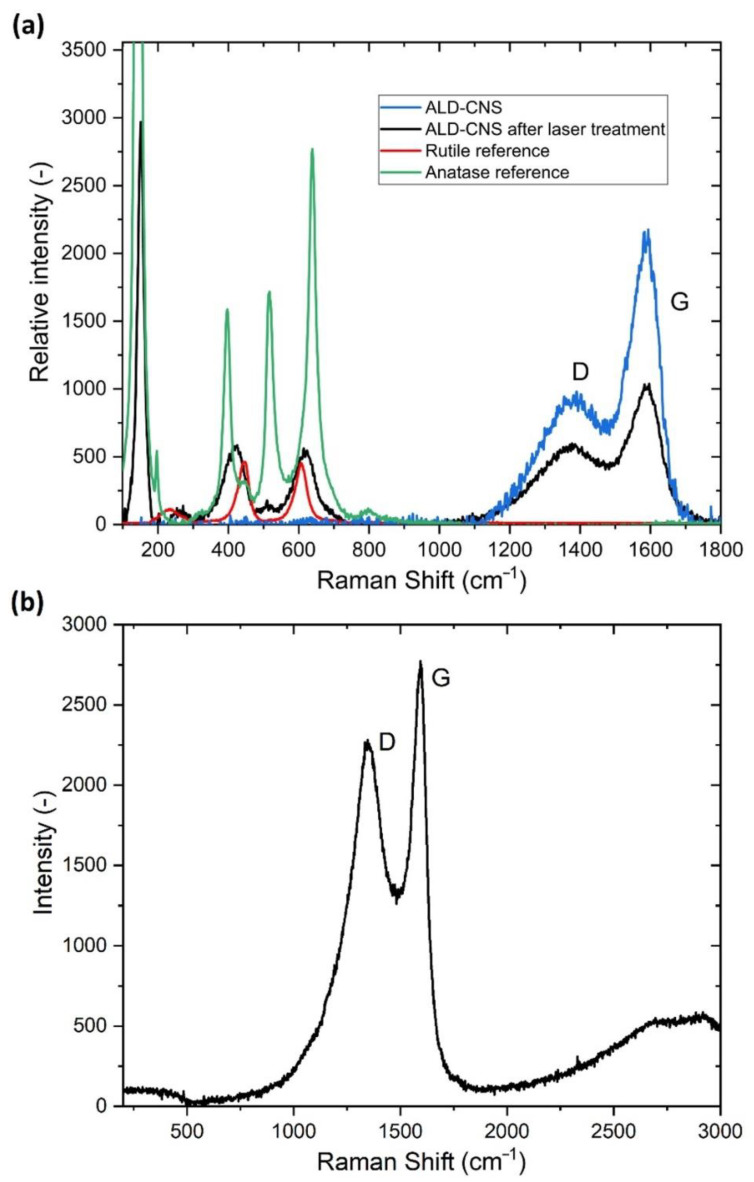
Raman spectra of (**a**) ALD-CNS with reference materials and (**b**) ALD-CNP.

**Figure 4 nanomaterials-12-02226-f004:**
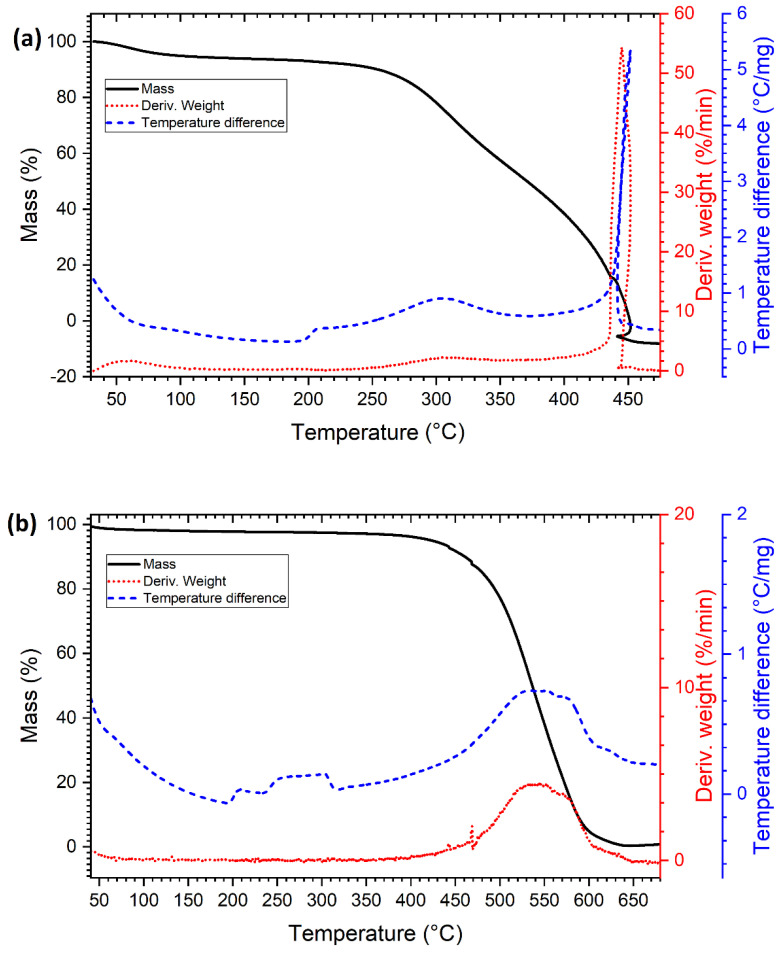
Thermal analysis curves for (**a**) ALD-CNS and (**b**) ALD-CNP nanoparticles in an air atmosphere.

**Figure 5 nanomaterials-12-02226-f005:**
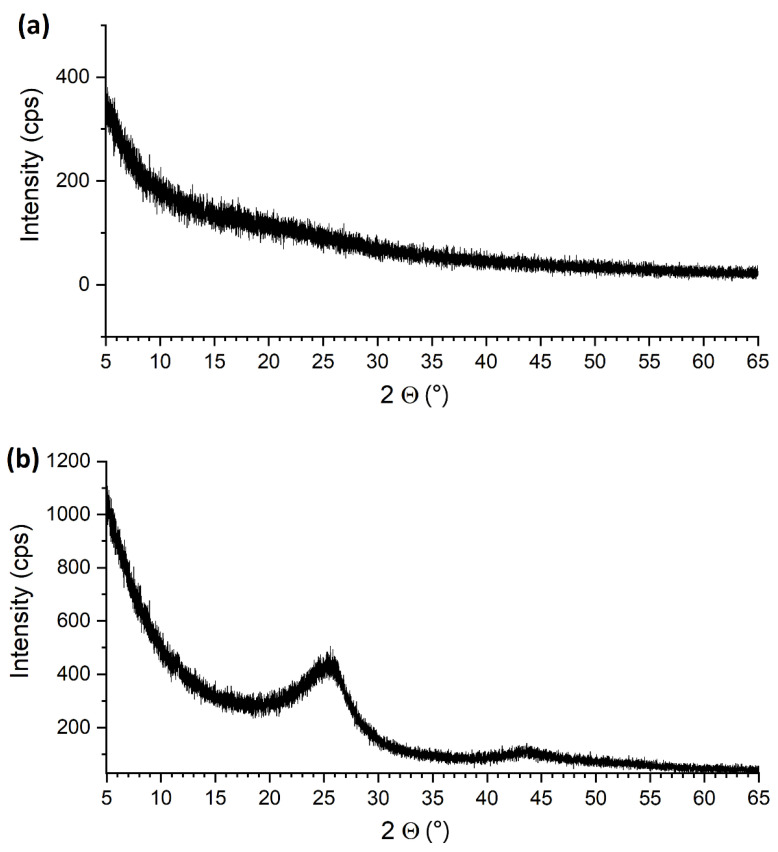
XRD pattern of (**a**) ALD-CNS and (**b**) ALD-CNP.

**Figure 6 nanomaterials-12-02226-f006:**
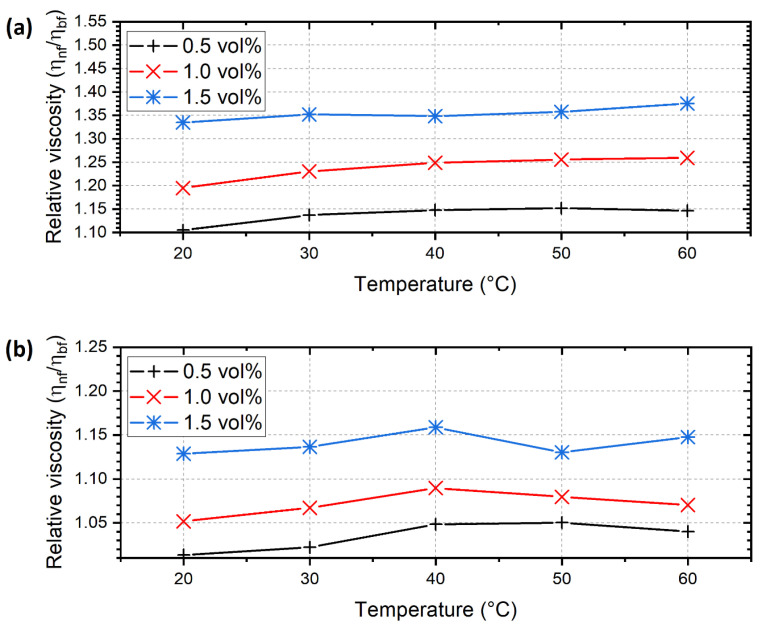
Relative viscosity of the (**a**) ALD-CNS and (**b**) ALD-CNP nanofluids (the base fluid for (**a**) is 1/5 ethanol–DI, and that for (**b**) is ethylene–glycol).

**Figure 7 nanomaterials-12-02226-f007:**
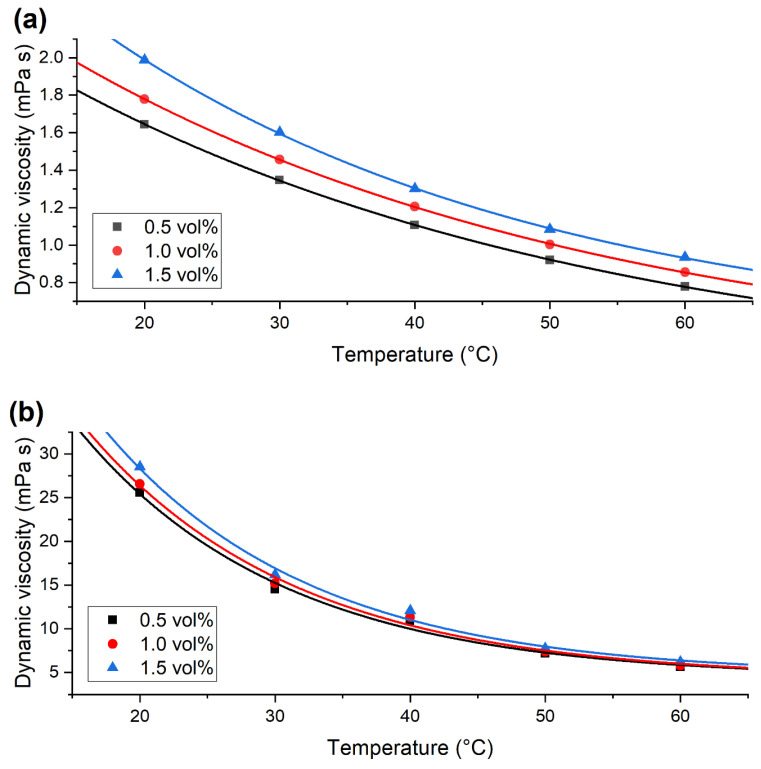
Dynamic viscosity of (**a**) ALD-CNS and (**b**) ALD-CNP nanofluids with a fitted exponential curve.

**Figure 8 nanomaterials-12-02226-f008:**
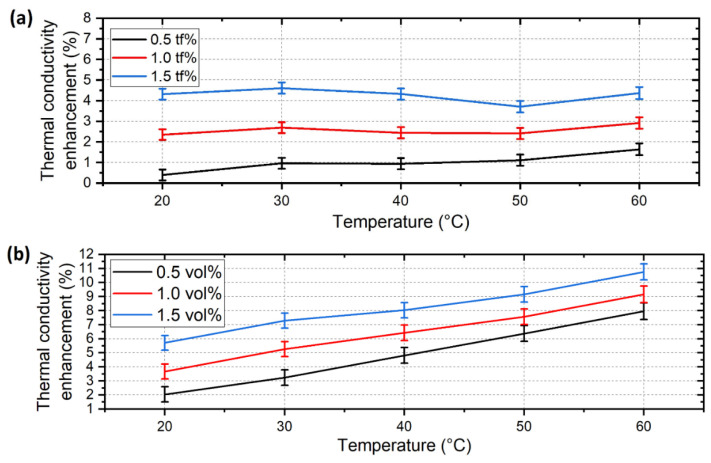
Thermal conductivity enhancement of (**a**) 1/5 ethanol-DI-based ALD-CNS and (**b**) ethylene–glycol-based ALD-CNP nanofluids.

**Figure 9 nanomaterials-12-02226-f009:**
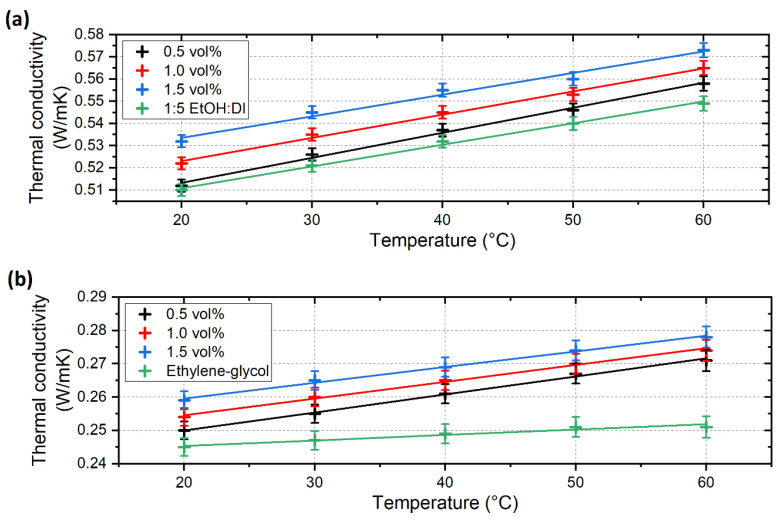
Absolute thermal conductivity of (**a**) ALD-CNS and (**b**) ALD-CNP nanofluids compared to their base fluids with linear fitted lines.

**Table 1 nanomaterials-12-02226-t001:** Fitting parameters for viscosity prediction by an exponential curve fit.

Nanofluid	A	B	C	R^2^
ALD-CNS 0.5 vol%	2.27	41.05	0.251	0.99992
ALD-CNS 1.0 vol%	2.41	40.10	0.313	0.99991
ALD-CNS 1.5 vol%	2.76	32.87	0.486	0.99974
ALD CNP 0.5 vol%	78.6	15.18	4.36	0.98868
ALD CNP 1.0 vol%	80.10	15.51	4.34	0.98921
ALD CNP 1.5 vol%	88.19	15.19	4.69	0.98902

**Table 2 nanomaterials-12-02226-t002:** Fitting parameters of thermal conductivity prediction.

Fluid	Equation	R^2^
ALD-CNS 0.5 vol%	λ = 1.12 × 10^−3^ °C + 0.491	0.99276
ALD-CNS 1.0 vol%	λ = 1.04 × 10^−3^ °C + 0.502	0.99175
ALD-CNS 1.5 vol%	λ = 9.73 × 10^−4^ °C + 0.514	0.97576
1:5 EtOH:Water	λ = 9.76 × 10^−4^ °C + 0.491	0.99386
ALD-CNP 0.5 vol%	λ = 5.42 × 10^−4^ °C + 0.239	0.99491
ALD-CNP 1.0 vol%	λ = 5.04 × 10^−4^ °C + 0.244	0.99368
ALD-CNP 1.5 vol%	λ = 4.73 × 10^−4^ °C + 0.250	0.99302
Ethylene–glycol	λ = 1.64 × 10^−4^ °C + 0.242	0.92986

**Table 3 nanomaterials-12-02226-t003:** Comparison of maximal thermal conductivity and viscosity increase between the current research and literature reports.

Ref.	Nanofluid	Thermal Conductivity Increase (%)	Relative Viscosity Increase (%)
-	ALD-CNS (1.5 vol%)	4.6	37.5
-	ALD-CNP (1.5 vol%)	10.8	15.9
[6]	CuO (5 vol%)	60	N/A
[6]	Al_2_O_3_ (5 vol%)	40	N/A
[7]	Copper (0.3 vol%)	40	N/A
[9]	TiO_2_ (2.0 vol%)	7.2	15
[10]	MgO (2.3 vol%)	17	68
[11]	TiO_2_ (1.0 vol%)	14.4	N/A
[11]	Al_2_O_3_ (1.0 vol%)	4	N/A
[11]	Fe (0.3 vol%)	16.5	N/A
[11]	WO_3_ (0.3 vol%)	13.8	N/A
[14]	MWCNT (0.5 vol%)	31.99	900
[14]	MWCNT (2.5 vol%)	69.68	2000
[20]	30:70 Graphene-TiO_2_ (0.5 vol%)	27.84	N/A
[21]	Gr-MWCNT/Cu	41	N/A

## Data Availability

The data presented in this study are available on request from the corresponding authors.

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
