# Peer review of "Thermal Conductivity Enhancement of Atomic Layer Deposition Surface-Modified Carbon Nanosphere and Carbon Nanopowder Nanofluids"

_nanomaterials, 2022, doi:10.3390/nano12132226_

Round 1

Reviewer 1 Report

The introduction needs to be rewritten. There are many energy conservation processes that do not involve heat exchange. Further, solar collectors can be also based on electromagnetic radiation usage, which does not rely on heat absorption. The authors refer specifically to solar thermal collectors.

Several references cannot be checked because of "Error! Reference source not found".

There are several language errors. Please fix them. For example, the first sentence of the abstract is not complete. Or the first sentence of the introduction: "... devices that are serve our modern life."

Author Response

Please refer to the uploaded document for the answers.

Reviewer 2 Report

In this study, the authors are conducting the experimental evaluation of thermal conductivity and viscosity of ALD-CNS and ALD-CNP nanofluids. They conclude that additional TiO2 layers drastically improve the thermal conductivity of ALD-CNS compared with that of bare CNS. It is believed that the present results are important and attract attention of researchers. However, there are some ambiguities and questions as below, which must be clarified before acceptance.

1. Figures are not linked well in the main text.

2. The words “et. al.” should be “et al.”

3. Lines 262-264: Why is there a difference in the polarity of the zeta-potential of ALD-CNS and ALD-CNP nanofluids? Why are the zeta-potentials negative for the ALD-CNS nanofluids and positive for the ALD-CNP nanofluids? Do the zeta-potential mean the electrical potential of the slip plane of the particle surfaces? Why can you prove the stability of the nanofluids by the zeta-potentials?

4. Lines 304-306 and 319-320: although the authors emphasize that the ALD-CNP is capable higher thermal conductivity enhancement on higher temperatures, while the ALD-CNS nanofluids had better results on lower temperatures, the temperature ranges preferable for ALD-CNP and ALD-CNS should be described quantitatively. It is not clear what you mean by the higher or lower temperatures.

--

Author Response

(The authors gave the same response as above.)

Author Response

(The authors gave the same response as above.)

Reviewer 4 Report

There are some comments for authors to improve the manuscript. Authors must make a major revision before publication.

1-The introduction is too poor. Authors should review one reference by one referenceand some main conclusions should be given. And authors should compare the references in the introduction.

2-What is the difference and innovation of this manuscript? Authors should explain the difference and innovation at the end of the introduction.

3-Errors are displayed in some places, please check the document. (In Error! Reference source not found. )

4-We all know that the thermal conductivity and viscosity change with temperature, and the author needs to fit a formula

5- The analysis section of this manuscript are too weak. Authors must make a revision on this, and deep mechanism analysis must be presented.

6-The stability of nanofluids is very important for nanofluids, and the authors did not study the stability of nanofluids.

7- Error analysis and experimental verification are absent, and the accuracy of results cannot be guaranteed.

8- The variation law of viscosity with temperature needs to be given.

9- The latest references are few, more latest references on nanofluids should be cited in the introduction.

Author Response

(The authors gave the same response as above.)

Round 2

Reviewer 1 Report

The authors improved the manuscript and applied my comments.

Reviewer 4 Report

Authors have revised the manuscript well and it can be accepted now.